# Capsule Endoscopy in Crohn’s Disease—From a Relative Contraindication to Habitual Monitoring Tool

**DOI:** 10.3390/diagnostics11101737

**Published:** 2021-09-22

**Authors:** Adi Lahat, Ido Veisman

**Affiliations:** 1Chaim Sheba Medical Center, Department of Gastroenterology, Sackler Medical School, Tel Aviv University, Tel Hashomer 52620, Israel; idoweiss37@gmail.com; 2Sackler Medical School, Tel Aviv University, Tel Aviv 67011, Israel

**Keywords:** capsule endoscopy (CE), Crohn’s disease (CD), inflammatory bowel disease (IBD), pan-enteric capsule, guidelines, capsule retention

## Abstract

Crohn’s disease (CD) is a chronic inflammatory disorder that may involve the gastrointestinal tract from the mouth to the anus. Habitual disease monitoring is highly important during disease management, aiming to identify and treat disease exacerbations, in order to avoid immediate and future complications. Currently, ilio-clonoscopy is the gold standard for mucosal assessment. However, the procedure is invasive, involves sedation and allows for visualization of the colon and only a small part of the terminal ileum, while most of the small bowel is not visualized. Since CD may involve the whole length of the small bowel, the disease extent might be underestimated. Capsule endoscopy (CE) provides a technology that can screen the entire bowel in a non-invasive procedure, with minimal side effects. In recent years, this technique has gained in popularity for CD evaluation and monitoring. When CE was first introduced, two decades ago, the fear of possible capsule retention in the narrowed inflamed bowel lumen limited its use in CD patients, and a known CD located at the small bowel was even regarded as a relative contraindication for capsule examination. However, at present, as experience using CE in CD patients has accumulated, this procedure has become one of the accepted tools for disease diagnosis and monitoring. In our current review, we summarize the historic change in the indications and contraindications for the usage of capsule endoscopy for the evaluation of CD, and discuss international recommendations regarding CE’s role in CD diagnosis and monitoring.

## 1. Introduction

Crohn’s disease (CD) is a chronic inflammatory disorder that may involve the gastrointestinal tract from the mouth to the anus. Disease location varies among patients, with small intestine involvement in approximately 75% of patients, and exclusive small bowel disease in 30% [1,2,3]. Inflammation is transmural, and might, therefore, be complexed by various intra-abdominal complications, such as fistula, abscess, perforations and fibrotic strictures [4]. The disease might produce significant morbidity and diminished quality of life (QOL) [5,6,7,8,9,10]. Disease behavior is characterized by episodes of exacerbations with symptom aggravation and periods of disease remission [11]. Habitual disease monitoring is highly important during disease management, aiming to identify and treat disease exacerbations in order to avoid immediate and future complications [12]. Monitoring the results has a direct influence on therapeutic decisions [12,13]. Monitoring is based upon symptoms and clinical evaluation, biomarker assessment, endoscopic scoring and imaging results. Naturally, the best monitoring should integrate all of the above; however, there are no consensus recommendations regarding optimal timing and monitoring tools at present [14]. 

The first goal in the treatment of a CD patient is to rapidly achieve clinical remission, and then to sustain this remission for a long period of time. Other goals include the prevention of disease progression, irreversible structural damage and medical complications, avoidance of hospitalization and surgeries and maintaining patient’s QOL [11,15]. Achieving these therapeutic goals include reaching biochemical and endoscopic remission, and, ideally, accomplishing deep remission, which is defined by the combination of clinical, biochemical, endoscopic, histological and imaging remission [16,17,18]. As endoscopic recurrence was shown to indicate future biochemical and clinical exacerbations, special attention is now paid to endoscopic surveillance [19,20]. At present, ilio-clonoscopy is the gold standard for mucosal assessment [21]. This allows for direct visualization of the diseased mucosa and enables biopsies and pathological testing to be performed. However, the procedure is invasive, involves sedation and allows for visualization of the colon and only a small part of the terminal ileum, while most of the small bowel is not visualized. Since CD may involve the whole length of the small bowel, disease extent might be underestimated.

There are few biochemical markers that might be used as surrogate markers for mucosal inflammation; the most studied are the inflammatory markers C-Reactive Protein (CRP, a marker for systemic inflammation) and calprotectin (specific marker for intestinal inflammation). However, these markers do not have a full correlation with endoscopic findings. Hence, up to one third of CD patients do not have elevated CRP during disease exacerbation [22,23], and calprotectin levels may not correlate with isolated ileal disease [24,25]. Bowel imaging studies may also play an important role in CD monitoring.

These serve in both the emergency setup and in habitual disease follow-up. In emergency cases, bowel imaging is used to diagnose intra-abdominal complications such as bowel perforation, fistulas, or abscess. During routine disease, monitoring bowel imaging can assess inflammation and intestinal damage, and affect treatment strategy. The accepted methods are computed tomography enterography (CTE), magnetic resonance enterography (MRE), and small bowel ultrasound. Each imaging technique has its specific advantages and drawbacks, yet none of them reach the accuracy and sensitivity of endoscopy in the assessment of mucosal inflammation [26].

Capsule endoscopy (CE) offers the technology needed to screen the entire bowel in a non-invasive procedure, with minimal side effects. In recent years, this technique has gained popularity for CD evaluation and monitoring [27,28]. In our current review, we summarize the historical change in indications and contraindications for the usage of capsule endoscopy for the evaluation of CD in the last two decades, and discuss international recommendations regarding CE’s role in CD diagnosis and monitoring. 

## 2. CE—The Early Days

Before the era of video capsule endoscopy, most of the small bowel was considered as the “blind spot” of the digestive system. As upper endoscopy could reach the duodenum, and lower endoscopy could reach the terminal ileum, most of the small bowel was left outside the reach of direct visualization. Push enteroscopy, which was introduced in the mid-nineties [29], could reach further down the small bowel; however, it still could not cover the small bowel to its full extent. Imaging modalities such as small bowel follow-through (SBFT) and, later, CT enteroscopy were used for small-bowel evaluation [30]. 

The first idea for direct visualization of the small bowel using military missile technology emerged in 1998 [31], and the first capsule endoscopy received the approval of the US Food and Drug Administration (FDA) in 2001. In the two decades that followed the FDA’s approval, the technology gained in popularity and became a useful diagnostic tool for various small-bowel pathologies. Current indications for CE include: advanced evaluation of obscure gastrointestinal (GI) bleeding (overt or occult), diagnosis of celiac disease and evaluation of refractory celiac disease, detections of malignancy of the small bowel, diagnosis of CD and evaluation of CD activity [32]. However, regarding the latter, the use of CE raised doubts in its early days. The main concern was the potentially serious adverse event of capsule impaction in the narrowed small-bowel lumen of CD patients.

Although CE displayed an innovative technology with a potentially major role in the management of CD, a technology which allows for a direct visualization of the small bowel mucosa and does not include exposure to radiation, patients with CD-suggestive symptoms or with signs of subacute obstructive symptoms were excluded from early clinical protocols [33,34,35].

Obscure GI bleeding was the first indication to be tested, and real-world studies showed an added value of 50–67% for CE compared to other modalities. In 2003, Saurin et al., demonstrated the superiority of CE in comparison with push-enteroscopy in patients with obscure GI bleeding [36,37,38]. In a short period of time, CE attained a dominant diagnostic role in the management of patients suffering from obscure GI bleeding. A few patients with CD were inadvertently included in these studies, and it transpired that the reason for GI bleeding was occult CD [33,34,37]. Thus, as data accumulated, and despite the initial apprehension, the use of CE for the diagnosis and follow-up of CD patients became widespread.

A few years after CE became available, several studies exemplified that CE outperforms computed tomography (CT) and enteoclysis (small-bowel enema). A comparative study from 2002 demonstrated a significant advantage in the use of CE over barium follow-through in 20 patients with suspected small-bowel disease. Barium follow-through was performed 4 days before CE was inserted in all patients. In this study, Barium follow-through was considered normal in 17 patients and, in 3 patients, showed ileal nodularity. However, CE showed pathologic findings in 17 patients and was normal in only 3 patients [39].

In 2003, Voderholzer and colleagues conducted a study comparing CE and CT enteroclysis in 22 patients with suspected small-bowel pathology. In this study, CE detected more small-bowel lesions than CT enteroclysis in patients with obscure GI bleeding and CD [40]. Thus, soon after the introduction of CE, and despite the justified fear of capsule retention in the potential narrowed lumen of the small bowel in CD patients, CE was shown to be superior to the accepted imaging modalities for the detection and diagnosis of small-bowel CD.

## 3. Current Data

As shown above, CD can affect the entire GI tract. Notably, up to 75% of CD patients will suffer from small bowel involvement [1]. At present, VCE may serve as an important tool for establishing CD diagnosis. However, in most cases, it is not the first-line measurement [41]. Hence, VCE usually serves as a diagnostic tool when ileocolonoscopy and radiology are negative or inconclusive in patients with significant clinical suspicion for small-bowel disease, which, in most cases, includes elevated inflammatory markers [42,43].

## 4. Diagnostic Yield of VCE

CD diagnosis is based on clinical features and endoscopic evaluation. The diagnosis is confirmed by histology results. The first step in the diagnosis of CD is to identify patients that suffer from characteristic symptoms such as chronic diarrhea and abdominal pain, combined with laboratory results pointing towards a chronic inflammatory condition. The next step during patient evaluation includes performing ileocolonoscopy with mucosal biopsies from the inflamed mucosa in order to establish the diagnosis [20]. However, a considerable proportion of CD patients may suffer from proximal disease, which is located in the small intestine, beyond the reach of the endoscope. Therefore, in these groups of patients, an endoscopic examination may not lead to diagnosis [44].

Data comparing the diagnostic yield of VCE for small-bowel CD with imaging modalities such as small-bowel barium follow-through (SBFT) and CTE show a clear superiority to VCE. In a meta-analysis by Choi et al., the diagnostic yield of VCE was superior to SBFT—66% vs. 21.3% [45]. Even when compared to the endoscopic procedure of push enteroscopy, VCE showed a significantly higher diagnostic yield [46,47]. Dionisio et al., evaluated the diagnostic yield of VCE compared to other noninvasive modalities including small bowel radiography, ileocolonoscopy, push enteroscopy, CTE, and MRE. In this meta-analysis, for suspected CD patients, VCE was superior in comparison to small bowel radiography, CTE, and iliocolonoscopy (52% vs. 16%, 68% vs. 21%, 47% vs. 25%, respectively) [47]. A retrospective study by González-Suárez and colleagues compared the diagnostic yield of VCE and MRE for the assessment of CD in patients with suspected or known small-bowel involvement. The results of this study demonstrated a significantly higher sensitivity of VCE in detecting proximal and distal disease in the small bowel (jejunum and ileum) compared to MRE (76.6% vs. 44.7% *p* = 0.001) [48].These differences are attributed to the high sensitivity of VCE to minor erosions or defects in the bowel mucosa- changes that are under the detection threshold of imaging modalities, and a high sensitivity to the length coverage of the small bowel, compared to the partial small-bowel visualization that occurs during endoscopy. 

## 5. CD Extent and Disease Activity Evaluation

After the diagnosis of CD, the disease extent throughout the GI tract should be established [49]. The current practice is to use MRE which allows for a transmural visualization of the small bowel and does not expose the patient to ionizing radiation and its future potential complications or involve invasive procedures [50]. However, VCE may identify small-bowel lesions that MRI might fail to detect. Although most guidelines have not recommended performing VCE in patients with normal MRE or CTE [18], it may be considered for certain indications, such as unexplained anemia, severe malnutrition, and inconsistency between symptoms and other imaging findings [41].

VCE may also serve as an important tool for monitoring disease activity, identifying patients at high risk for disease relapse, and assessing response to treatment. All these capsule capacities can play a major role in the management of CD patients [11]. Currently, the habitual follow-up strategies for CD patients include clinical evaluation, use of biomarkers, endoscopy, imaging including MRE, CTE, small-bowel ultrasound, and VCE [51]. Regarding the latter, the experience gained over the last two decades has demonstrated that VCE is an efficient and safe technology. Kopylov et al., evaluated the clinical impact of VCE in patients with established CD. In this cross-sectional study, VCE provided meaningful results, leading to therapeutic changes in more than 50% of CD patients [52]. Similar results were demonstrated by Dussault et al., In this retrospective study, among the 71 CD patients who underwent VCE, 38 patients experienced a change in their treatment regimen due to severe lesions that were found during VCE examination [53]. Another study by Kim et al., showed similar results; in this study, therapeutic strategies were modified in 70.2% and 50% of patients with suspected CD and established CD, respectively [54]. According to a meta-analysis by Yung and colleagues, VCE may serve as a sensitive modality in the detection of post-operative disease recurrence in CD patients, similar to MRE and small bowel US. In this systemic review and meta-analysis, the pooled sensitivity for VCE was 100%, with a specificity of 69% for assessing postoperative endoscopic recurrence in CD patients [51].

## 6. Adverse Events

In terms of adverse events, the main possible risk of performing VCE is capsule retention, which could potentially lead to small-bowel obstruction. Even without obstruction, retention could lead to endoscopic or surgical intervention. This risk is increased in patients with fibrostenotic CD or with a history of small-bowel obstruction. Capsule retention is defined as the presence of the capsule endoscope in the gastrointestinal tract for a minimum of 2 weeks [55]. The retention risk among healthy patients is negligible, but the risk is increased among CD patients. In a recent study by Pasha et al., the retention rate among patients with suspected CD was 2.35%, and the retention rate in established CD was 4.63% [56]. However, the use of a specific patency capsule, which is radio-opaque and dissimilates in the bowel in the case of retention prior to VCE in a high-risk population, significantly decreases the retention rate and is advisable for CD patients with structuring disease. In a recent study, MRE for the prediction of capsule retention in patients with established CD had a sensitivity of 92% and specificity of 59%. Hence, this study demonstrated the importance of the patency capsule, since, if MRE was the reference for a decision on CE performance, at least 40% of patients would not have performed the examination [57].

Incomplete small-bowel examination may occur, due to the limited battery life, in up to 12% of cases. This mainly affects the older population and patients with alerted bowel motility. In this group of patients, the use of pro-kinetic agents (such as metoclopramide or erythromycin) is advisable [58].

Another important consideration, which mainly affects pan-enteric capsule endoscopy (PCE), is bowel preparation. While standard preparation for small bowel VCE usually includes the ingestion of only clear fluids for 24 h prior to the procedure and a 12-h overnight fast [59], for colon preparation, a 4 L split-dose polyethylene glycol (PEG) preparation is used [59]. An additional bolus with picosulfate (Pico-Salax^®^, Ferring, Germany) is given after the capsule reaches the small bowel to facilitate small-bowel transit, and another bolus is given 2 h after this. Naturally, as strong laxatives, all these preparation medications cause diarrhea and abdominal discomfort. Thus, while preparation for small-bowel CE is relatively patient-friendly, the preparation for PCE is very demanding and challenging, especially for CD patients that suffer from various abdominal symptoms. In a recent study by our group assessing patients’ preference for diagnostic procedures, small-bowel VCE was widely accepted by the patients and preferred over MRE due to its having fewer side effects, but patients that underwent colonic preparation reported severe discomfort during preparation [59]. These patients did not show a significant preference for CE over MRE. 

Despite the benefits of VCE and the growing usage of this modality in CD patients, the interpretation of study results still needs standardization. To date, there is no definitive diagnostic criterion for CD. A few different CE scores for CD have been offered and validated in recent years. The most commonly used to date is the Lewis score (LS), which uses an algorithm that separates the small bowel into three parts and designates points to different disease characteristic findings, including strictures, ulcers, or fistulas, in each of the segments [59]. Capsule Endoscopy Crohn’s Disease Activity Index (CECDAI) is an additional score utilized to assess small-bowel inflammation. This score divides the small bowel into two segments—proximal and distal—and includes the degree and extent of mucosal inflammation and the presence of strictures [60]. Yablecovitch and colleges demonstrated that both systems are equally suitable for assessing mucosal inflammation in established CD patients [61]. 

A new CD-specific VCE score, the Eliakim score [62], was recently published and validated using the pan enteric CD capsule. The new score was designed and validated to measure disease activity index using a pan enteric capsule, and has the advantage of visualization the full length of the bowel mucosa. However, to date, there is no gold-standard scoring system for the diagnosis of CD with VCE.

## 7. Technical Advancements 

Since VCE was first introduced, it has undergone many technical changes and developments, including a faster frame rate and enhanced image resolution, leading to higher efficacy. In 2006, a colon capsule was introduced with the ability to visualize the colon’s full length. Recently, the PillCam Crohn’s Capsule, a pan-enteric video capsule system, was introduced, which enables visualization of the small and the large bowel [63]. This capsule was specifically developed according to the unmet need of a full-length visualization of the digestive system to accurately assess disease involvement throughout the entire bowel mucosa. The PCE was designed with an adaptive frame rate (AFR) technology, which adjusts the rate of image capture to 4–35 FPS, depending on the movement speed, and has a 336-degree view via two camera heads, which enables visualization of the small bowel and colon. The CD-specific, AI-applied software enables a comparison of images and disease progression over time, a comparison of images to existing catalog images, classification of severity by location and an assessment of mucosal healing.

Many other capsule types (e.g., Jinshan CE, MiroCam CE, Olympus ENDOCAPSULE EC-10, CapsoVision) exist, with diverse technical capabilities. However, since this review’s scope is CD, we chose to focus on the specific CD PillCam Crohn’s Capsule.

The recently published Eliakim score (see above) [62] uses this pan enteric capsule (PCE) for a full disease activity index.

PillCam Crohn’s Capsule, a pan-enteric video capsule, is shown in Figure 1. Figure 2 and Figure 3 demonstrate typical CD lesions in the small bowel detected by the CD capsule.

## 8. Current International Recommendations

As the usage of CE in CD has become the common practice, a few international recommendations have been issued in recent years regarding its utility in the diagnosis and management of CD [14,17,18,42,64,65]. 

These international recommendations focus on several key topics that reflect diverse aspects of the treatment of CD. Most consensus recommendations reflect two major topics: appropriate selection of patients and sagacious use of CE during disease monitoring and treatment. Herein, we will review both topics. 

## 9. Appropriate Patients’ Selection

The first topic accessed in the international guidelines is appropriate patient selection. As for all diagnostic procedures, targeting patients that will benefit from the procedure while avoiding potential complications as much as possible is a major goal. 

Thus, the current recommendations identify a few potential candidates: patients with known or suspected widespread disease that affects the small and the large intestine will unquestionably benefit from a single procedure using PCE. PCE enables the identification and monitoring of mucosal involvement throughout the digestive tract, with high sensitivity and specifity [66,67]. 

Another important potential benefit of CE is the ability to monitor and guide therapeutic decisions in patients with known small-bowel disease. Consequently, patients with known disease, including post-surgical patients that are candidates for initiating or changing current treatment, are good candidates for CE [14,17,18,42,64,65]. 

Pediatric patients are another potential candidate for CE. Endoscopic procedures in these patients are usually performed under general anesthesia, and CE can spare these patients from undergoing an invasive procedure under general anesthesia [68]. 

Naturally, appropriate patient selection includes avoiding complications as much as possible. Therefore, CE is contraindicated in patients with known gastrointestinal obstructions unless intestinal patency is proven, usually with the patency capsule [14,17,18,42,64,65,68]. 

## 10. Disease Monitoring and Treatment

The ECCO guidelines for IBD diagnosis and management, issued in 2019, state that: “Endoscopic or transmural response to therapy should be evaluated within 6 months following initiation of therapy [14]” and that: “Endoscopic or cross- sectional reassessment in CD should be considered in cases of relapse, persistent disease activity, new unexplained symptoms, and prior to switch of therapy” [14]. The same guidelines emphasize the superiority of CE compared to other imaging modalities to determine small-bowel disease activity [69] and in the assessment of mucosal healing [60,70,71]. 

A recent study assessing patients’ compliance with and preference for MRE versus CE showed a clear benefit of CE in terms of discomfort and side effects [72]. Taken together, the superiority VCE showed in assessing disease activity and in patients’ preference favor the selection of CE when choosing between CE and MRE for long-term follow- up 14. Furthermore, the introduction of the new pan enteric CE [28] facilitates disease mapping and grading in a single examination. The examination is relatively safe and does not require sedation, thus enabling more frequent monitoring. VCE also plays a role in postoperative CD patients. Several studies have demonstrated that VCE increases the diagnostic accuracy and might greatly impact therapeutic decisions [73] and, according to the AGA guidelines, patients with suspected disease recurrence, which is undiagnosed by ileocolonoscopy or imaging studies [65].

## 11. CE and COVID 19

The World Health Organization declared a worldwide pandemic due to the coronavirus disease 19 (COVID-19) on 11 March 2020 [74].

Though human-to-human transmission occurs mainly through air droplets or direct contact, other pathways such as environmental contamination, fecal–oral transmission, and fomites were also identified [75]. Therefore, both upper and lower endoscopy are considered high-risk procedures for the entire endoscopic team. While upper endoscopic procedures expose the endoscopic personnel to pulmonary and gastric secretions, lower endoscopy procedures expose the team to fecal remnants containing unknown viral inoculum [76]. 

As a response to the COVID-19 outbreak, dynamic regulations were issued worldwide, social distancing was recommended and specific recommendations for safe endoscopic practice were published [77,78,79].

Together with strict personnel protection, invasive procedures were limited [77,78,79,80,81]. Furthermore, patients were reluctant to arrive at medical centers due to fear of exposure to the virus [81,82]. In order to avoid unnecessary clinic visits, various strategies, such as remote medicine (Telecare, Telemedicine), became highly popular [83]. Capsule endoscopy, from this perspective, may provide complete bowel screening with minimal clinic presence time (only the time needed to swallow the capsule and upload the program), and with no invasive procedure needed [82].

## 12. CE and deep Learning Neural Network

Automated image analysis is termed computer vision, which is a multi-disciplinary field that focuses on computers’ gaining knowledge of digital images [84]. In recent years, artificial intelligence (AI) deep learning algorithms, termed convolutional neural networks (CNNs), have reformed the computer vision field, offering significant accuracy in various image analysis fields, including medical image analysis [84,85,86].

CE has a preferable imaging modality for AI-based analysis given its dependence on configuration recognition in still figures. 

Current CE reading and interpretation by a single reader is very time-consuming, even when performed by an experienced reader. Automatic pathology detection using AI might benefit from pathology detection while reducing the necessitated reading time [87].

A recent meta-analysis [87] found that the pooled sensitivities and specificities for ulcer detection in 19 AI studies performed on CE interpretations were 0.95 (95% confidence interval ((CI), 0.89–0.98) and 0.94 (95% CI, 0.90–0.96), respectively. The pooled sensitivities and specificities for bleeding or bleeding source found in this meta-analysis were 0.98 (95% CI, 0.96–0.99) and 0.99 (95% CI, 0.97–0.99), respectively.

Specific algorithms for CD detection in CE showed high accuracy levels.

For the detection of CD ulcers by randomly split images, the area under the curve (AUC) was 0.99, and accuracies ranged from 95.4% to 96.7%. For individual patient-level experiments, the AUCs were 0.94–0.99 [88]. For the differentiation of CD-related strictures, the results were also impressive. For classifying strictures, the accuracy was 93.5% (±6.7%). For the differentiation of strictures from normal mucosa area, the AUC was 0.989; for differentiation between strictures and all ulcers, the AUC was 0.942; for the differentiation between strictures and different ulcer grades, the AUC was 0.89–0.99 [89].

Hence, AI uses a preferred and highly effective technique to improve CE’s interpretation accuracy and minimize reading time. Future improvements to the algorithm’s technology will further affect its utility in CE, and will facilitate the use of CE in CD. 

## 13. Cost Effectiveness of CE in CD

The cost-effectiveness of using CE as a disease-monitoring tool in CD was recently evaluated in two different studies. Naturally, cost-effectiveness is affected by local economic health care expenses, and, therefore, is specific to an individual country. 

A recent study assessing cost-effectiveness in the United States (US) [90], evaluated patients’ quality of life (QOL) and the costs of CE compared to the common monitoring practices of Ilecocolonoscopy and small bowel imaging (CTE or MRE), with regard to disease severity. The results showed improved QOL in all degrees of disease severity, with the greatest improvement seen in active symptomatic patients. In a 20-year period, CE reduced expenses ($313,367 vs. $320,015), increased life expectancy (18.15 vs. 17.9 years) and increased QOL (8.7 versus 8.0 QALY). Thus, CE was obviously the cost-effective alternative.

Using CE was shown to decrease the costs of monitoring and surgery and increase the costs of treatment. CE was cost-effective in 71% of individual patients and 78% of populations.

Another recent study, assessing cost-effectiveness and patient outcome in the British National Health Service (NHS), reached similar results [91]. In a 20-year period, total cost per patient was reduced by using CE. (£38,043 vs. £42,266). Similar to the findings from the US, during the first 2 years, the costs per CE were higher due to the higher treatment expenses. Thereafter, costs were reduced due to a decrease in the number of surgeries. QOL was 10.96 vs. 10.67 QALY, in favor of CE. CE was cost-effective for 74% of patients.

Therefore, looking at cost-effectiveness and QOL, CE is superior to colonoscopy+ imaging modality, and should be regarded as the preferred alternative when feasible. 

## 14. Conclusions

Since the introduction of VCE, 20 years ago, as a new and accurate tool for diagnosing and assessing small-bowel lesions, many new indications have emerged. One of the most-studied indications is CD. At first, the concerns regarding capsule retention limited its usage in CD patients, and the first cases were patients evaluated for other common indications (e.g., iron deficiency anemia) with the diagnosis of CD as the cause of the symptom. As experience and data accumulated, the use of CE in the diagnosis and management of CD became prevalent, and at present this procedure appears in all national and international guidelines as a major tool in diagnosis and monitoring throughout habitual disease management. Over the years, technical improvements such as better visual abilities, the development of the patency capsule, the colon capsule, and the pan enteric capsule further facilitated and enhanced its robust position in the diagnosis and management of CD. Recently, AI options were added to the software and facilitated accurate and timely readings. Furthermore, the cost-effectiveness of CE usage during CD monitoring was recently demonstrated. As technology advances, and with the recognition and growing popularity of AI, we are sure that the use of VCE in IBD management and research will increase in the coming years.

The manuscript, including the related data, has not been previously published and is not under consideration elsewhere

## Figures and Tables

**Figure 1 diagnostics-11-01737-f001:**
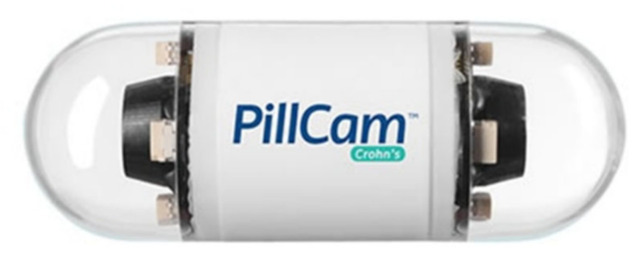
Pan enteric CD capsule.

**Figure 2 diagnostics-11-01737-f002:**
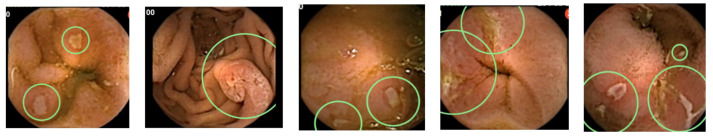
CE images of typical CD-related small-bowel ulcerations. Lesions are circled.

**Figure 3 diagnostics-11-01737-f003:**
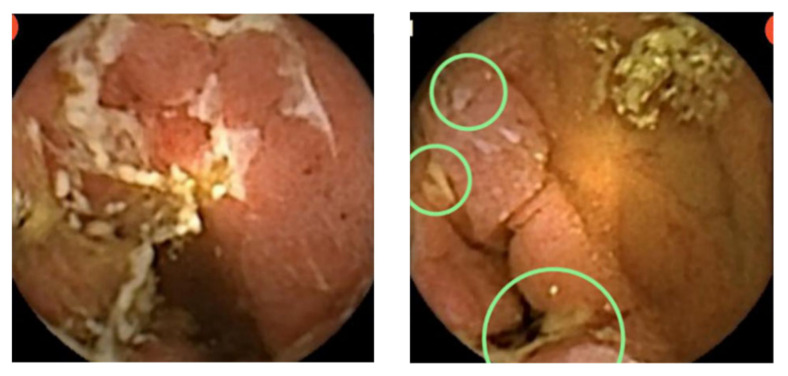
CE images of severe small bowel CD—deep ulcerations and extended disease, including lumen narrowing. The green circles mark pathological findings consistent with CD-ulcers and erosions.

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
