# Peer review of "Capsule Endoscopy in Crohn’s Disease—From a Relative Contraindication to Habitual Monitoring Tool"

_diagnostics, 2021, doi:10.3390/diagnostics11101737_

Round 1
Reviewer 1 Report
In this paper, the authors present the review on the Capsule Endoscopy (CE). They summarize historically the change in indications and contraindications for the usage of capsule endoscopy for the evaluation of Crohn’s disease (CD) in the last two decades, and discuss international recommendations regarding CE role during CD diagnosis and monitoring.
The topic of this paper is significant and the contents are of reference value. The paper organizations are acceptable.
Some suggestions:
- The abstract should be simplified and more clearly presented since it is too tedious.
- When some abbreviated words are first used, such as CT, MR, it is better to give full words.
- Representation of section 3 to 5 could be more concise.
- In section “Technical advancements”, it is better to compare other types of Capsule Endoscope products with the PillCam.
- Some contents could be better illustrated by using figures and tables.
Author Response
- The abstract should be simplified and more clearly presented since it is too tedious.
We thank the reviewer for his/her comment, the abstract was corrected and shortened accordingly.
- When some abbreviated words are first used, such as CT, MR, it is better to give full words.
We thank the reviewer for his/her comment. The initials were written in their full form when they first appeared in the text.
- Representation of section 3 to 5 could be more concise.
We thank the reviewer for his/her comment and modified this accordingly.
- In section “Technical advancements”, it is better to compare other types of Capsule Endoscope products with the PillCam.
We thank the reviewer for his/her comment. More capsule types including illustrations were added to the text at the Technical advancements section.
- Some contents could be better illustrated by using figures and tables.
We thank the reviewer for his/her comment, more Images were added to the text.

Reviewer 2 Report
Comments
In this article, the authors detail the role of CE in Crohn’s disease
- Abstract written well. Few grammatical errors noted which are minor (abstract last but third line-need a comma, introduction 5th line-inappropriate spacing and also few sections in the main manuscript).
- The introduction, the the role of CE in the early days, its progress for evaluation of the CD is appropriately mentioned.
- Few things need consideration: CE needs a bowel prep is a significant limitation for patient with symptomatic CD. There should be a additional information on this section and methods to decrease the failed study. This should be inaddition to battery issues/ fibrostenosed CD with retained capsule
- Section on role of the CE during the COVID-19 pandemic given the decreased utlization of the endoscopy and its potential role to increase its use when patients are not able to get the invasive procedures should be highlighted (please refer to PMID: 32281689, PMID: 33162736).
- Section on deep learning neural networks (/ordina) use for expansion of CE role in CD needs to be mentioned
- Please expand the role of CE in postoperative management of the CD as its need is emerging in this space.
- Finally the cost effectiveness needs to be mentioned.
Author Response
Reviewer 2:
- Abstract written well. Few grammatical errors noted which are minor (abstract last but third line-need a comma, introduction 5th line-inappropriate spacing and also few sections in the main manuscript).
- The introduction, the the role of CE in the early days, its progress for evaluation of the CD is appropriately mentioned.
- Few things need consideration:
CE needs a bowel prep is a significant limitation for patient with symptomatic CD. There should be a additional information on this section and methods to decrease the failed study. This should be inaddition to battery issues/ fibrostenosed CD with retained capsule
We thank the reviewer for his/her comment, a full section discussing bowel preparation and patients’ compliance was added to the text at paragraph 6- adverse events.
- Section on role of the CE during the COVID-19 pandemic given the decreased utlization of the endoscopy and its potential role to increase its use when patients are not able to get the invasive procedures should be highlighted (please refer to PMID: 32281689, PMID: 33162736).
- We thank the reviewer for his/her comment, a new paragraph discussing CE at the times of COVID 19 pandemic was added to the text (paragraph 12).
- Section on deep learning neural networks (/ordina) use for expansion of CE role in CD needs to be mentioned
We thank the reviewer for his/her comment, a new paragraph discussing AI in CE was added to the text (paragraph 13).
- Please expand the role of CE in postoperative management of the CD as its need is emerging in this space.
We thank the reviewer for his comment, data and more references to this topic were added under section 11 – “Disease monitoring and treatment”.
- Finally the cost effectiveness needs to be mentioned.
We thank the reviewer for his/her comment, a new paragraph cost effectiveness of CE was added to the text (paragraph 14).

Round 2
Reviewer 1 Report
This is the revised version, and it is acceptable.
Author Response
Thank you for your revision